# Coping Style and Resilience Mediate the Effect of Childhood Maltreatment on Mental Health Symptomology

**DOI:** 10.3390/children9081118

**Published:** 2022-07-27

**Authors:** Hua Cao, Ruiqi Zhang, Ling Li, Ling Yang

**Affiliations:** Key Laboratory of Behavioral and Mental Health Symptomology of Gansu Province, School of Psychology, Northwest Normal University, Lanzhou 730070, China; caohua360083915@163.com (H.C.); liling9092@126.com (L.L.); yangling@nwnu.edu.cn (L.Y.)

**Keywords:** childhood maltreatment, coping style, resilience, mental health symptomology

## Abstract

Background: A well-known distal risk factor for mental health symptomology is childhood maltreatment. Previous research revealed that several mediators, such as coping style and resilience, might be connected to the psychological mechanism of childhood maltreatment on mental health symptomology. Objective: The purpose of this study was to assess how coping style and resilience affect the relationship between childhood maltreatment and mental health symptomology of college students. Methods: With the method of cross-sectional survey, 740 college students from China (Gansu Province) completed the Childhood Trauma Questionnaire (CTQ), the Simplified Coping Style Questionnaire (SCSQ), the Connor–Davidson Resilience Scale (CD-RISC), and the Symptom Checklist 90 (SCL-90). Structural equation modeling (SEM) was used to reveal the link between childhood maltreatment, coping style, resilience, and mental health symptomology. Results: The results showe that childhood maltreatment was significantly positively correlated with mental health symptomology and significantly negatively correlated with coping style and resilience. Coping style was significantly negatively correlated with mental health symptomology and significantly positively correlated with resilience. Resilience was significantly negatively correlated with mental health symptomology. Coping style and resilience played a partially mediating role in the relationship between childhood maltreatment and college students’ mental health symptomology. Through a chain of intermediary effects on coping style and resilience, childhood maltreatment not only had a direct impact on mental health symptomology but also had an indirect impact. Conclusion: Childhood maltreatment could affect college students’ mental health symptomology through the chain mediating effect of coping style and resilience. Therefore, it is an effective way to reduce the influence of childhood maltreatment on mental health symptomology through some intervention measures to cultivate positive coping style and improve resilience.

## 1. Introduction

Health refers not only to the healthy physical development of an individual, but more importantly, to a stable and positive state of mental health symptomology for a long time. The term “mental health” is used to describe a person’s mental health status, ranging from positive mental health to poor mental health. The former refers to that all aspects and activities of psychology are in a good or normal state, while the latter is usually called mental health problems [1]. The China National Mental Health Development Report (2019–2020) revealed that the mental health index of the 18~24-year-old group is lower than that of all other age groups, which indicated that mental health problems were more prevalent among college students. Numerous investigations had revealed that interpersonal communication [2], academic pressure [3], and parenting style [4] were the main factors leading to the mental health problems among college students, and these factors were all distributed in their daily life. It can be seen that the surroundings play an important role in college students’ mental health problems [5].

Bronfenbrenner’s Ecological Systems Theory states that individual development is influenced by several environmental systems, of which the distal chronological system (the past experience of individual) is one of the factors affecting the development of individuals [6]. Moreover, psychoanalytic theory also holds that the early experiences of individuals is very important, and that the misery or happiness that individuals encounter in their lives can be traced back to their past experiences, especially the early childhood experience. A study of 668 college students found that college students who experienced negative traumatic events in their early adulthood, experience an impact on their physical and mental health and social life as adults [7]. Childhood maltreatment is conceptualized in terms of neglect (physical and/or emotional) and abuse (physical, emotional, and/or sexual) [8]. Childhood maltreatment predicted depression [9], anxiety [10], aggression and violence [11], self-harm behavior [12], and substance abuse in adolescents [13]. The above findings and theories confirmed that the experiences of childhood maltreatment led to increased mental health problems [14]. However, not all college students who experienced adversity can develop mental health problems; instead, some adolescents develop a positive coping style and eventually overcome their difficulties [15]. Coping style refers to the cognitive process in which individuals deal with stress and thus restore internal and external balance. Additionally, the positive consequences obtained by individuals resisting adversity are defined as resilience. Therefore, if resilience is an outcome, then coping style is the specific process of the role of resilience in the process of physical and psychological recovery of the individual [16]. The relationship between coping style and resilience was inextricably linked from a stress-coping perspective [17].

Coping style is defined as the thought processes and behaviors that a person sometimes engages in when he or she becomes aware of the need to manage or prevent harm and loss. One study found that childhood maltreatment was significantly and positively associated with negative coping style, and individuals who did not experienced childhood maltreatment were more likely to choose a positive coping style [18]. Folkman and others argued that it was not the stress itself that affected an individual’s physical and mental health but the way they cope with it [19]. Furthermore, coping style was considered to be a protective factor for the physical and mental health of individuals [20]. Some studies showed that an immature or negative coping style led to psychological abnormalities among students [21], while a positive coping style was beneficial for mental health and reduced symptoms of depression and anxiety [22]. Coping theory emphasizes the cognitive and behavioral efforts made by individuals in the face of stress, which influence the nature and intensity of the stress response during stressful events, and thus regulate the relationship between stress and individuals’ mental health [23]. A survey of African American women found that the symptoms of psychological problems were affected by coping style after the experience of maltreatment [24]. Gökmen’s study also found that coping style acted as a mediator that helps mitigate the effects of maltreatment experiences on the mental health symptomatology of adolescents [25]. Thus, we hypothesized that coping style mediated the relationship between childhood maltreatment and mental health symptomatology.

Resilience was understood as a person’s ability to always solve problems when encountering difficulties in different areas of life and to increase one’s resistance to future stress and trauma in the process of overcoming difficulties [26]. The post-traumatic resilience development model theory proposed by AGAIBI et al., (2005) emphasizes that when an individual experiences traumatic events in their early years, he or she activates dynamic coping mechanisms to protect himself or herself [27], including cognitive processes about the self and the external world, mobilization and use of protective factors, and personal personality traits. Resilience may be a protective factor that mediates between childhood maltreatment and mental health symptomatology and may prevent the development of psychiatric symptoms in individuals who have experienced maltreatment [28]. One study found that emotional abuse, emotional neglect, and sexual abuse were negatively associated with resilience [29]. Individuals with higher levels of resilience reduced the occurrence of their mental health symptomatology [30]. In exploring the impact of resilience on the mental health symptomatology of adolescents with experiences of childhood maltreatment, Fritz et al., (2018) found that individuals with higher levels of resilience were better able to adapt themselves in the face of stress, thereby promoting their mental health development [31]. More and more studies had found that resilience alleviated the depressive symptoms in individuals experiencing abuse and neglect [28]. Moreover, a recent study also found that resilience mediated the relationship between childhood maltreatment and college students’ mental health symptomatology [32]. Zhao et al., (2021) also found that individuals who adopt a positive coping style were more willing to use a variety of social resources to solve problems [3], so they had higher levels of resilience and better mental health [33]. The above studies suggest that coping style and resilience play a role in the relationship between childhood maltreatment and mental health symptomatology among college students. However, most of the previous research tends to focus on a single or individual factor. At an integrative level, the mechanism by which the interaction of coping style and resilience influenced the relationship between childhood maltreatment and mental health symptomatology remains poorly researched and discussed.

Based on the above findings and theories, this study aimed to investigate the effects of college students’ childhood maltreatment on mental health symptomology and explored the role of coping style and resilience in the relationship between childhood maltreatment and college students’ mental health symptomology, so as to provide evidence for preventing and intervening in college students’ mental health problems. The hypothesis of this study is that: (1) childhood maltreatment of college students positively predicts mental health symptomology, and (2) coping style and resilience play a chain mediating role in the relationship between childhood maltreatment and mental health symptomology, as shown in Figure 1. Specifically, childhood maltreatment caused college students to be more inclined to adopt negative coping tendencies, and then their resilience level decreased, which ultimately affected their mental health symptomology.

## 2. Methods

### 2.1. Study Design and Participants

The data for this cross-sectional study were collected from a convenience sample of college students from China (Gansu Province). The participants were recruited during the second term of 2021. Of the 820 students invited to participate in the survey, 810 returned the questionnaires. The other 70 questionnaires were excluded because 10 or more questions in a row received the same answer, the answers were presented in a particular pattern (such as 121212, etc.), and 5 or more questions were missed. In total, the sample was of 740 college students from freshman to seniors, with an effective rate of 91.36%. There are 233 boys (31.5%) and 507 girls (68.5%). In total, 440 freshmen (59.5%), 82 sophomores (11.1%), 147 juniors (19.9%), and 71 seniors (9.6%). All participants provided written informed consent prior to the study and voluntarily participated in the survey, and participants under the age of 18 were given written consent by their parents. All of the data on the main variables of interest (i.e., Childhood Maltreatment, Coping Style, Resilience, and Mental Health Symptomology) used in this study were collected on the same day. Information such as gender, grade, and nature of school was collected through demographic surveys. The questionnaires were completed in a quiet classroom environment and took about 15–25 min. This study was approved by the ethics committee of Northwest Normal University School.

### 2.2. Instruments

#### 2.2.1. Childhood Maltreatment

The Childhood Trauma Questionnaire (CTQ) was used to investigate the frequency of abuse and neglect before 16 years old. The CTQ is a 28-item retrospective self-report inventory divided into five dimensions: emotional abuse, physical abuse, sexual abuse, emotional neglect, and physical neglect [34,35]. Each item is scored on a 5-point Likert-type scale, with a total score of 25–125. The total score of CTQ was the sum of 25 items except three valid items. The higher the score, the more traumatic childhood experiences the individual is likely to have. The Chinese version of the childhood maltreatment questionnaire showed good psychometric properties in adolescents [36]. In this study, the Cronbach’s α coefficient was 0.93.

#### 2.2.2. Coping Style

The coping style was assessed with the Simplified Coping Style Questionnaire (SCSQ) [37]. It is a 20-question self-report scale divided into two dimensions: positive coping style and negative coping style. It is used to assess a person’s coping style when dealing with things. These questions are rated on a scale from 1 (never) to 4 (always). The total score of SCSQ is the positive dimension standard score minus the negative dimension standard score. The higher the score, the more likely the individual is to use positive coping styles. The Chinese version of the Simple Coping Style Questionnaire showed good reliability and validity among college students [38]. In this study, the Cronbach’s α coefficient was 0.93, and the Cronbach’s α coefficients of positive coping and negative coping were 0.94 and 0.85, respectively.

#### 2.2.3. Resilience

The Conner–Davidson Resilience Scale (CD-RISC) was used to evaluate psychological resilience related to coping skills on stressful events [39]. The CD-RISC is a 25-item self-report divided into three dimensions: tenacity, strength, and optimism and is used to assess a person’s ability to deal with stress and adversity. Each question is scored by 5-point Likert-type scale, ranging from 0 (completely incorrect) to 4 (correct almost all the time). The questions are summed up to obtain a total score. The higher the score, the higher the individual’s resilience level. The Chinese version of the Resilience Scale also showed good reliability and validity [40]. In this study, the Cronbach’s α coefficient of this scale was 0.93.

#### 2.2.4. Mental Health Symptomology

The Symptom ChecklList 90 (SCL-90) was used to assess individual psychological symptoms. The SCL-90 is a 90-item self-report scale containing ten dimensions of somatization, obsessive–compulsive, interpersonal sensitivity, depression, anxiety, hostility, phobic anxiety, paranoid ideation, and psychoticism [41]. Each question is scored by 5-point Likert-type scale, and the scores of these questions range from 1 (from none) to 5 (serious). The total score of SCL-90 is the sum of the 90 items. The higher the score, the more mental health problems the individual has. The Chinese version of SCL-90 showed good reliability and validity among college students [42]. In this study, the Cronbach’s α coefficient of this scale was 0.97. The split-half reliability test shows that the Cronbach’s α was 0.93.

### 2.3. Control Variables

We controlled the gender of the participants (0 = male; 1 =female) because it was reported that gender was related to the mental health of individuals [43].

### 2.4. Data Analysis

In this study, the data obtained were entered and organized using SPSS22.0, and the hypotheses were verified by structural equation modeling using Amos24.0. First, because all the data in this study come from the self-report of the subjects, there is a possibility of a common method bias effect [44]. So a common method bias test was conducted. Generally, the first factor with an eigen root value greater than one explaining less than 40% of the total variance was considered to be free of common method bias effects. Descriptive statistics (mean and standard deviation) were used to describe the characteristics of college students, and an independent sample *t* test was used to compare the gender differences of CTQ, SCSQ, CD-RISC, and SCL-90. Cohen’s d is used to estimate the effect quantity [45]. Additionally, correlation analysis was used to test the correlation between CTQ, SCSQ, CD-RISC, and SCL-90.

Second, the hypothetical model (Figure 1) was tested using Amos by building a structural equation model. The structural equation model was able to validate the relationship among childhood maltreatment, coping style, resilience, and mental health symptomology. Latent factor ‘childhood maltreatment’ was represented by five indicator variables; ‘mental health symptomology’ was represented by nine indicator variables; ‘coping style’ was represented by two indicator variables; and ‘resilience’ was represented by three indicator variables in the hypothesized model. The hypothesized model was tested by fitting the model using individual path coefficients and fit indicators, comparing the indicators, and conducting chi-square/ratio of freedom tests. The modelfit was determined based on the comparativefit index (CFI) and the Tucker–Lewis index (TLI) (CFI and TLI values ≥0.9 are indicative of acceptable modelfit), and the Root Mean Square Error of Approximation (RMSEA; good modelfitted indicated by values <0.08) [46]. Indirect associations were examined to understand whether: (i) coping style mediated the association between childhood maltreatment and mental health symptomology; (ii) resilience mediated the associations between childhood maltreatment and mental health symptomology; and (iii) coping style and resilience mediated the association between childhood maltreatment and mental health symptomology. The significance of the hypothesized model mediation effects was tested by estimating 95% confidence intervals (CIs) for the mediation effects for a bootstrap sample of 5000. If the 95% CI did not contain 0, the effect was indicated to be significant.

To calculate statistical power for a moderated mediation model with current sample size (*n* = 740), a series of Carlo power analyses with 10,000 replications using maximum likelihood estimation was conducted in Mplus version 8.3 to determine the power for α = 0.05. Monte Carlo power analyses suggested that the sample size was sufficiently large to detect small effects (i.e., 0.20) in a multiple mediation model with power >0.80 [47].

## 3. Results

### 3.1. Common Method Deviation Test

The Harman single factor test can minimize the bias effect of the common method. For all items, the principal component factor analysis without rotation was carried out. It was found that there were six factors with eigen root values greater than 1, and the first factor explained 36.53% of the total variance, which is less than 40% of the criterion. Therefore, the problem of common method deviation does not exist in this study.

### 3.2. Descriptive Statistics

The independent sample *t* test was used to analyze the difference between gender and each dimension of the four variables. The results show that there are gender differences in physical abuse, emotional abuse, and neglect (*t* = 4.49, *p* < 0.001; *t* = 2.67, *p* = 0.008; *t* = 4.43, *p* < 0.001); boys scored higher than girls. There are also gender differences in the dimensions of depression, anxiety, hostility, and fear (*t* = −2.08, *p* = 0.038; *t* = −2.19, *p* = 0.029; *t* = 3.64, *p* < 0.001; *t* = −2.92, *p* = 0.004), in which the scores of girls with depression, anxiety, and fear are higher than those of boys. As in Table 1.

### 3.3. Mechanism of Childhood Maltreatment Affecting Mental Health Symptomology

#### 3.3.1. Correlation Analysis of Childhood Maltreatment, Coping Style, Resilience, and Mental Health Symptomology

The correlation analysis of each research variable is shown in Table 2. There was a significant negative correlation between coping style and childhood maltreatment (β = −0.492, 95% CI [−0.584, −0.388]). Additionally, there was a significant positive correlation between resilience and childhood maltreatment (β = 0.655, 95% CI [0.534, 0.749]), and a significant negative correlation between childhood maltreatment and coping style and resilience (β = −0.159, 95% CI [−0.316, −0.031]). Mental health symptomology was negatively correlated with resilience (β = −0.238, 95% CI [−0.408, −0.067]). There was a significant positive correlation between childhood maltreatment and mental health symptomology (β = 0.463, 95% CI [0.357, 0.578]).

#### 3.3.2. Chain Intermediary Test of Coping Style and Resilience

The Amos structural equation model was used for the test, and the fitting results show that the fitting indexes of x²/df = 4.767, NFI = 0.934, IFI = 0.947, CFI = 0.947, TLI = 0.933, and RFI = 0.916 are all above 0.9; Additionally, RMSEA = 0.071 is less than the critical value of 0.08, so the fitting degree of the model is good (Figure 2).

The confidence intervals corresponding to both the direct effect and the three paths of this model test did not contain 0, indicating that the effects were significant, and the chain mediation was established. See Table 2. In the final model, the direct path from childhood maltreatment to mental health symptomology was also significant (β = 0.463, *p* < 0.001, 95% CI [0.357, 0.578]). The indirect effect of childhood maltreatment on mental health symptomology was not only through coping style (β = −0.268, *p* < 0.001, 95% CI [−0.469, −0.104]) but also mediated by resilience (β = 0.089, *p* = 0.006, 95% CI [0.030, 0.184]). The indirect effect of childhood maltreatment on mental health symptomology by coping style and resilience: β = 0.180, *p* < 0.006, 95% CI [0.052, 0.342]. Positive coping style and resilience masked the consequences of childhood maltreatment on mental health symptomology, and childhood maltreatment had a strong indirect impact on depression (see Table 2).

## 4. Discussion

To the best of our knowledge, this was the first study to investigate the mediating roles of coping style and/or resilience on the relationship between childhood maltreatment and mental health symptomology in a sample of college students in order to examine the underlying mechanisms of childhood maltreatment on mental health symptomology. This study suggested that childhood maltreatment may impair coping style and reduce resilience, both of which then contribute to the problem of poor mental health. It also confirmed the direct detrimental effect of childhood maltreatment on mental health symptomology.

It is consistent with previous studies [48]; childhood maltreatment positively predicted college students’ mental health symptomology, indicating that college students who had adverse experiences in childhood had a higher chance of having mental health problems. This may be due to the lack of communication between college students with childhood maltreatment experiences and their caregivers in childhood, not feeling the care and love they deserved, and not forming correct and healthy value orientations during their growth process, resulting in their failure to form a healthy psychology [49]. The studies support that trauma could predict the development of psychological problems.

The findings of this study indicate that resilience and coping style, respectively, partially mediated the relationship between childhood maltreatment and the mental health symptomology of college students. First of all, childhood maltreatment negatively predicted coping style, while coping style positively predicted college students’ mental health symptomology [50]. This suggests that college students who had experienced childhood maltreatment are more likely to adopt negative coping strategies when faced with difficulties. When children were confronted with abuse and neglect, their internal evaluations were usually negative, manifested as attempts to cope with it by some ways such as self-blame, fantasy, and retreat to temporarily relieve or avoid pain [51]. If they experienced more abuse and neglect in childhood, encountering similar stressful events in adulthood would also reignite the negative coping style from childhood maltreatment, which was continuously negatively reinforced and in turn led to a range of psychological problems in individuals [52].

Secondly, similar with other research findings, childhood maltreatment was significantly and adversely linked with resilience, and resilience positively predicted the mental health symptomology of college students [53]. It showed that the more serious the childhood maltreatment college students experienced, the less conducive the development of college students’ psychological quality was, which then affected the mental health symptomology of college students. The resilience protection model holds that individuals with higher resilience are more conducive to the establishment and development of their positive psychological quality [27]. One of the important protection factors for individuals to cope with setbacks and adversity was high resilience. When negative events happen, high resilience people also have fewer problems than others [54]. When a negative event occurs, individuals with higher resilience have a lighter definition of the severity of the event, resulting in a lower psychological burden and a lower probability of psychological problems.

In this study, the chain mediation effects of childhood abuse on the mental health symptomology of college students were first through coping style and then through resilience. The results show that childhood maltreatment negatively predicted coping style, coping style positively predicted resilience, and resilience positively predicted college students’ mental health symptomology. This was consistent with the research results of Yang et al. [55]. In this study, coping style and resilience did not fully mediate the relationship between childhood maltreatment and mental health symptomology. This may be due to the dysfunction of the hypothalamus–pituitary–adrenal axis in individuals with childhood maltreatment experience [56], decreased amygdala [57], and the inability of the prefrontal cortex to control the amygdala normally [58]. These abnormalities are not only the neurobiological characteristics of individual psychological problems [59], but also increase the probability of individual suffering from psychological problems [60]. Numerous research revealed a strong relationship between coping strategy and resilience [61], which was supported by the findings of this study. Coping style is one of the important internal protection factors of individual resilience [62], and individuals who adopted a positive coping style were more likely to choose to face the problem and solve it and were also more willing to actively seek help from society and others [63]. In other words, when experiencing negative events, individuals who adopted a positive coping style actively mobilized their own internal and external protection factors to resist the current difficulties and pressures and took a face-to-face approach or good emotional control to alleviate the impact brought by stress events [64], thus improving their level of resilience. Therefore, the more an individual adopted a positive coping style to deal with events in life, the higher the level of resilience [65,66]. It also suggested that individuals who took active coping styles could quickly adapt to the current unfavorable environment when facing the pressure from work, life, and study and actively seek for the favorable factors to solve the current problems and protect themselves from harm. The intervention of coping style was beneficial to the reduction of individual mental health problems and the cultivation of positive psychological quality [67], and coping style was an important influencing factor of individual resilience [68], thus improving the level of resilience and ultimately promoting individual mental health.

This study had several limitations: First, the cross-sectional study method adopted in this study did not fully explain the causal relationship among childhood maltreatment, coping style, resilience, and mental health symptomology. Future studies could use other approaches, such as longitudinal designs or experimental studies, to continue to explore this question in greater depth. Second, with the gradual expansion of the definition of childhood maltreatment, the five types of traumas assessed by CTQ self-report were no longer able to accurately assess the situation of childhood maltreatment. Therefore, in future research, a more comprehensive questionnaire or a combined questionnaire should be used to assess childhood maltreatment. Thirdly, this study only set six demographic variables, such as gender, only child, grade, learning nature, nationality, and family residence. However, the factors that actually affected college students’ mental health were complex, such as individual academic achievement, which had certain limitations in the scope of research. In future related research, we could make a more comprehensive and thorough analysis of the influencing factors and select different demographic data from different angles for research. Finally, while coping style and resilience did not fully mediate the relationship between childhood maltreatment and mental health symptomology, mechanisms of other mediating variables could be explored in future research. This study only focused on the effect of coping style and resilience on the relationship between childhood maltreatment and mental health symptomology.

## 5. Conclusions

The results of this study show significant correlations among childhood maltreatment, coping style, resilience, and mental health symptomology. We found that childhood maltreatment indirectly affected mental health symptomology through coping style and resilience, indicating that childhood maltreatment increased the choice of negative coping style, decreased resilience, and promoted mental health problems. This conclusion also provided two ideas for us to promote college students’ mental health. One was to address the problem at the source, that is, to reduce the occurrence of childhood maltreatment. Carry out publicity and education in the form of family activities for children’s caregivers, not only to pay attention to the physical development of children, but also to create a warm and safe family atmosphere and emotional support to give more attention to children and pay attention to the communication and exchange with children. Secondly, according to the results of this study, a series of activities were carried out to guide students’ coping style and improve their resilience. For example, the mental health week was used to play short animation videos about coping styles, and different coping styles bring different results to guide individuals to choose more appropriate coping styles when they encounter things. Another initiative is to develop a series of small activities centered on the ABC theory of emotion (trivia roleplay, etc.) to change irrational beliefs and improve resilience.

## Figures and Tables

**Figure 1 children-09-01118-f001:**
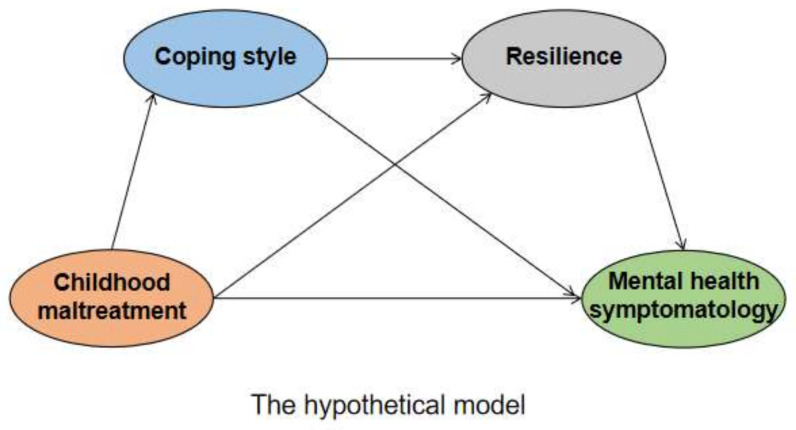
The hypothetical model: exploring the role of coping style and resilience between childhood maltreatment and mental health symptomology.

**Figure 2 children-09-01118-f002:**
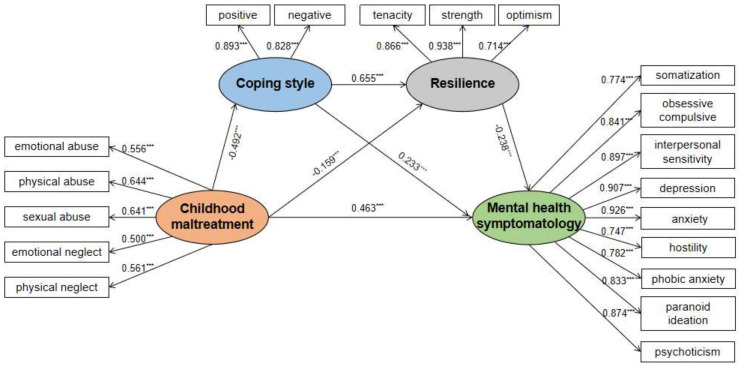
Final model: the influencing direction of coping style and resilience was the same compared to the hypothetical model. Note: *** means *p* < 0.001.

**Table 1 children-09-01118-t001:** College student’s demographic characteristics for the entire sample and divided by sex (*n* = 740).

Characteristics	*n*	All %/M ± SD	*n*	Girls %/M ± SD	*n*	Boys %/M ± SD	Sex Comparison t *p*-Value	
*Sex*								
Female	508	68.6						
Male	232	31.4						
*School grade*								0.104
Freshmen	439	59.3	287	56.5	152	65.5		
Sophomores	84	11.4	55	10.8	29	12.5		
Juniors	148	20	112	22.1	36	15.5		
Seniors	69	9.3	54	10.6	15	6.5		
*Childhood Maltreatment*								
Emotional abuse	740	7.33 ± 2.82	508	7.25 ± 2.81	232	7.51 ± 2.83	1.19	0.233
Physical abuse	740	6.09 ± 2.21	508	5.82 ± 1.97	232	6.67 ± 2.56	4.49	0.000
Sexual abuse	740	5.81 ± 2.01	508	5.72 ± 1.97	232	6.00 ± 2.09	1.75	0.082
Emotional neglect	740	11.29 ± 5.56	508	10.90 ± 5.45	232	12.11 ± 5.72	2.67	0.008
Physical neglect	740	8.77 ± 3.19	508	8.42 ± 3.15	232	9.53 ± 3.15	4.43	0.000
*Mental Health Symptomatology*								
Somatization	740	17.76 ± 5.42	508	17.84 ± 5.54	232	17.59 ± 5.17	−0.60	0.547
Obsessive–compulsive	740	20.06 ± 5.86	508	20.24 ± 5.87	232	19.66 ± 5.82	−1.25	0.211
Interpersonal sensitivity	740	16.56 ± 5.07	508	16.50 ± 5.09	232	16.70 ± 5.05	0.50	0.615
Depression	740	22.55 ± 6.89	508	22.88 ± 7.14	232	21.81 ± 6.26	−2.08	0.038
Anxiety	740	16.51 ± 5.03	508	16.77 ± 5.20	232	15.94 ± 4.57	−2.19	0.029
Hostility	740	9.75 ± 3.22	508	9.44 ± 2.97	232	10.43 ± 3.61	3.64	0.000
Phobic anxiety	740	11.11 ± 3.76	508	11.37 ± 3.91	232	10.55 ± 3.34	−2.92	0.004
Paranoid ideation	740	9.55 ± 2.94	508	9.43 ± 2.95	232	9.83 ± 2.92	1.72	0.086
Psychoticism	740	15.39 ± 4.61	508	15.26 ± 4.57	232	15.69 ± 4.70	1.18	0.240
*Coping Style*								
Positive coping style	740	2.71 ± 0.51	508	0.12 ± 0.97	232	−0.25 ± 1.01	−4.67	0.000
Negative coping style	740	2.25 ± 0.53	508	0.01 ± 1.01	232	−0.02 ± 0.97	−0.29	0.775
*Resilience*								
Tenacity	740	39.80 ± 8.48	508	39.89 ± 8.09	232	39.59 ± 9.28	−0.42	0.673
Strength	740	26.38 ± 5.40	508	26.59 ± 5.26	232	25.91 ± 5.68	−1.55	0.121
Optimism	740	11.72 ± 2.96	508	11.75 ± 2.94	232	11.66 ± 3.03	−0.37	0.713

Note: mean (M); standard deviation (SD).

**Table 2 children-09-01118-t002:** Standardized direct and indirect association coefficients of the final model.

Variables	β	*p*-Value	95% CI
Direct Associations			
*Coping style*			
Childhood maltreatment	**−0.492**	**0.001**	**−0.584, −0.388**
*Resilience*			
Coping style	**0.655**	**0.001**	**0.534, 0.749**
Childhood maltreatment	**−0.159**	**0.013**	**−0.316, −0.031**
*Mental health symptomatology*			
Resilience	**−0.238**	**0.007**	**−0.408, −0.067**
Coping style	0.233	0.179	0.093, 0.385
Childhood maltreatment	**0.463**	**0.001**	**0.357, 0.578**
Indirect Associations			
Childhood maltreatment–Coping style– Resilience–Mental health symptomatology	**0.180**	**0.006**	**0.052, 0.342**
Childhood maltreatment–Coping style–Mental health symptomatology	**−0.268**	**0.001**	**−0.469, −0.104**
Childhood maltreatment–Resilience–Mental health symptomatology	**0.089**	**0.006**	**0.030, 0.184**

Note: The greater the absolute value of β, the better the performance. Statistically significant (*p* < 0.05) associations are marked in bold.

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
