# Peer review of "Coping Style and Resilience Mediate the Effect of Childhood Maltreatment on Mental Health Symptomology"

_children, 2022, doi:10.3390/children9081118_

Round 1

Reviewer 1 Report

Dear Sir/Mam

Please find bellow the requested review regarding the manuscript. The article contains a lot of useful information on the issue. The topic is very interesting and use of sources is appropriate. In addition, it lacks tables that would be very useful and can contain critical information.

The article contains a lot of useful information on the issue. It is quite clear what is already known about this topic and the research question is clearly outlined. The abstract is too brief and Discussion section involve too much information. There must be a balance in the manuscript

Specifically

Introduction

Introduction section doesn’t involve too much information, while Conclusion section is too long. There is an asymmetry in the manuscript.

Research methods

Method is unclear. The authors must explain with more details.

Results

Results are unclear. The authors must explain with more details.

Tables with demographic and other information are necessary.

Positive: There are some strengths of the article that could have an impact in the field, such as the topic and its impact on the existed literature. The manuscript is approved after major changes.

Author Response

#Response to reviewer1 comments:

In addition, it lacks tables that would be very useful and can contain critical information.

Many thanks to the reviewers for the comments. We have added two tables, namely College student’s demographic characteristics for the entire sample and divided by sex (n = 740) and Standardised direct and indirect association coefficients of the final model.

In the Results section, we have added rich content and marked in bule.

The abstract is too brief and Discussion section involve too much information. There must be a balance in the manuscript.

Many thanks to the reviewers for the comments. We divided the abstract into five parts: background, objective, method, result and conclusion. The results and conclusions are revised as follows “Results: The results showed that childhood maltreatment was significantly positively correlated with mental health symptomology, and significantly negatively correlated with coping style and resilience. Coping style was significantly negatively correlated with mental health symptomology, and significantly positively correlated with resilience. Resilience was significantly negatively correlated mental health symptomology . Coping style and resilience played a partially mediating role in the relationship between childhood maltreatment and college students' mental health symptomology. Childhood maltreatment not only directly affected mental health symptoms, but also indirectly affected mental health symptomology through a chain intermediary affects of coping style and resilience. Conclusion: Childhood maltreatment could affect college students' mental health symptomology through the chain mediating effect of coping style and resilience. Therefore, it is an effective way to reduce the influence of childhood maltreatment on mental health symptomology through some intervention measures to cultivate positive coping style and improve resilience.”

In the Abstract section, we have added rich content and marked in blue.

Introduction section doesn’t involve too much information, while Conclusion section is too long. There is an asymmetry in the manuscript.

Thanks to the reviewers for the comments. We have reconstructed the introduction part. Firstly, there was an evidence part about the relationship between childhood maltreatment and mental health symptomology, and it was described that not everyone who experiences adversity will react in the same way. Then, we mentioned several theories in PTSD and resilience, suggesting that coping style can affect resilience. Redefine what coping style was, described the literature on how childhood maltreatment affects coping style , and research on coping style and mental health symptomology. Then, it has mentioned the gap in exploring the intermediary to deal with the relationship between childhood maltreatment and mental health symptomology. (Resilience is put forward with the same logic). The details are as follows: The above findings and theories confirmed that the experiences of childhood maltreatment led to increased mental health problems (Ainamani et al., 2021). However, not all college students who experienced adversity can develop mental health problems; instead, some adolescents develop positive coping style and eventually overcome their difficulties (Su et al., 2020). Coping style refers to the cognitive process in which individuals deal with stress and thus restore internal and external balance. And the positive consequences obtained by individuals resisting adversity are defined as resilience. Therefore, if resilience is an outcome, then coping style is the specific process of the role of resilience in the process of physical and psychological recovery of the individual(Sun YueYi, 2019). The relationship between coping style and resilience was inextricably linked from a stress-coping perspective(Clauss-Ehlers, 2008).

Coping style is defined as the thought processes and behaviors that a person sometimes engages in when he or she becomes aware of the need to manage or prevent harm and loss. One study found that childhood maltreatment was significantly and positively associated with negative coping style, and individuals who did not experienced childhood maltreatment were more likely to choose positive coping style (Zhou, 2016). Folkman and others argued that it was not the stress itself that affected an individual's physical and mental health, but the way they cope with it (Folkman et al., 1987). Furthermore, coping style was considered to be a protective factor for the physical and mental health of individuals(Tol et al., 2013). Some studies showed that immature or negative coping style led to psychological abnormalities among students (Pengju et al., 2018), while positive coping style was beneficial for mental health and reduce symptoms of depression and anxiety (Sun et al., 2019). Coping theory emphasized the cognitive and behavioral efforts made by individuals in the face of stress, which influence the nature and intensity of the stress response during stressful events, and thus regulate the relationship between stress and individual's mental health(Chi, 2005). A survey of African-American women found that the symptoms of psychological problems were affected by coping style after the experiences of maltreatment (Mills et al., 2018). Gökmen's study also found that coping style acted as a mediator that help mitigate the effects of maltreatment experiences on the mental health symptomatology of adolescents(Arslan, 2017). Thus, we hypothesized that coping style mediated the relationship between childhood maltreatment and mental health symptomatology.

Resilience was understood as a person's ability to always solve problems when encountering difficulties in different areas of life, and to increase one's resistance to future stress and trauma in the process of overcoming difficulties (Sisto et al., 2019). The post-traumatic resilience development model theory proposed by AGAIBI et al. (2005) emphasizes that when an individual experiences traumatic events in their early years, he or she will activate dynamic coping mechanisms to protect himself or herself, including cognitive processes about the self and the external world, mobilization and use of protective factors, and personal personality traits. Resilience may be a protective factor that mediates between childhood maltreatment and mental health symptomatology and may prevent the development of psychiatric symptoms in individuals who have experienced maltreatment (Lee et al., 2018). One study found that emotional abuse, emotional neglect, and sexual abuse were negatively associated with resilience (Kesebir et al., 2015). Individuals with higher levels of resilience reduced the occurrence of their mental health symptomatology (Xu et al., 2018). In exploring the impact of resilience on the mental health symptomatology of adolescents with experiences of childhood maltreatment, Fritz et al. (2018) found that individuals with higher levels of resilience were better able to adapt to themselves in the face of stress, thereby promoting their mental health development. More and more studies had found that resilience alleviated the depressive symptoms in individuals experiencing abuse and neglect (Lee et al., 2018). Moreover, a recent study also found that resilience mediated the relationship between childhood maltreatment and college students' mental health symptomatology (GAO et al., 2020). Zhao et al. (2021) also found that individuals who adopt positive coping style were more willing to use a variety of social resources to solve problems, so they had higher levels of resilience and better mental health (Ge et al., 2020). The above studies suggested that coping style and resilience played a role in the relationship between childhood maltreatment and mental health symptomatology among college students. However,most of the previous researches tended to focus on a single or individual factor. At an integrative level, the mechanism by which the interaction of coping style and resilience influenced the relationship between childhood maltreatment and mental health symptomatology remain poorly researched and discussed.

As it coincides with the suggestion of the third reviewer, in the Introduction, we have added rich content and marked in green.

Method is unclear. The authors must explain with more details.

Thanks to the reviewers for the comments. In the method part, we added the recruitment procedure of research participants, calculated the sample size of this study, provided research ethics information, and added gender as a control variable, etc.

The contents of the Study Design and Participants' part modification were: “ The data for this cross-sectional study were collected from a convenience sample of college students from China (Gansu Province). The participants were recruited during the second term of 2021. Of the 820 students invited to participate in the survey, 810 returned the questionnaires. In total, a sample of 740 college students from freshman to seniors, with an effective rate was 91.36%. All participants provided written informed consent prior to the study and voluntarily participated in the survey, and participants under the age of 18 were given written consent by their parents. All of the data on the main variables of interest (i.e., Childhood Maltreatment, Coping Style, Resilience and Mental Health Symptomology) used in this study were collected on the same day. Information such as gender, grade, nature of school is collected through demographic surveys. The questionnaires were completed in a quiet classroom environment and took about 15–25 min. This study has been approved by the ethics committee of Northwest Normal University School.”

The contents of the Control variables' part added were: “We controlled the gender of the participants (0 = male, 1 =female ) because it was reported that gender was related to the mental health of individuals (e.g., Cauce et al., 2000) . ”

The contents of the Data analysis' part added were: “ Latent factor‘childhood maltreatment’ was represented by five indicator variables; ‘mental health symptomology’ was represented by nine indicator variables; ‘coping style’ was represented by two indicator variables; ‘resilience’ was represented by three indicator variables in the hypothesised model. Indirect associations were examined to understand whether:(i) Coping style mediated the association between childhood maltreatment and mental health symptomology; (ii) Resilience mediated the associations between childhood maltreatment and mental health symptomology; and (iii) Coping style and resilience mediated the association between childhood maltreatment and mental health symptomology. To calculate statistical power for moderated mediation model with current sample size (N = 740), a series of Carlo power analyses with 1,0000 replications using maximum likelihood estimation was conducted in Mplus version 8.3 to determine the power for α=0.05. Monte Carlo power analyses suggested that the sample size was sufficiently large to detect small effects (i.e., 0.20, Funder & Ozer, 2019) in multiple mediation model with power > 0.80. ”

As it coincides with the suggestion of the first reviewer, in the Study Design and Participants, Control variables and Data analysis section, we have added rich content and marked in bule and red.

Results are unclear. The authors must explain with more details.

Thanks to the reviewers for the comments. We added some details to the Descriptive statistics and Mechanism of childhood maltreatment affecting mental health symptomology in the results, as follows: The independent sample t test was used to analyze the difference between gender and each dimension of the four variables. The results showed that there were gender differences in physical abuse, emotional abuse and neglect (t=4.49, p<0.001; t=2.67, p=0.008; T=4.43, p<0.001), boys scored higher than girls. There are also gender differences in the dimensions of depression, anxiety, hostility and fear (t=-2.08, p=0.038; t=-2.19, p=0.029; t=3.64, p<0.001; t=-2.92, p=0.004), in which the scores of girls with depression, anxiety and fear are higher than those of boys. As in Table 1. The correlation analysis of each research variable is shown in Table 2. There was a significant negative correlation between coping style and childhood maltreatment (β=-0.492, 95%CI [-0.584, -0.388]). And a significant positive correlation between resilience and childhood maltreatment (β=0.655, 95%CI [0.534, 0.749]), and a significant negative correlation between childhood maltreatment and coping style and resilience (β=-0.159, 95%CI [-0.316, -0.031]). Mental health symptomology was negatively correlated with resilience (β=-0.238, 95%CI [-0.408, -0.067]). There was a significant positive correlation between childhood maltreatment and mental health symptomology(β=0.463, 95%CI [0.357, 0.578]). In final Model, the direct path from childhood maltreatment to mental health symptomology was also significant (ß = 0.463, p < 0.001, 95%CI [0.357, 0.578]). The indirect effect of childhood maltreatment on mental health symptomology was not only through coping style (ß = -0.268, p <0.001, 95%CI [-0.469, -0.104]), but also mediated by resilience (ß = 0.089, p = 0.006, 95%CI [0.030, 0.184]). The indirect effect of childhood maltreatment on mental health symptomology by coping style and resilience (ß = 0.180, p < 0.006, 95%CI [0.052, 0.342]).

In the Results section, we have added rich content and marked in blue.

Tables with demographic and other information are necessary.

Many thanks to the reviewers for the comments. We have added two tables, namely College student’s demographic characteristics for the entire sample and divided by sex (n = 740) and Standardised direct and indirect association coefficients of the final model.

In the Results section, we have added rich content and marked in bule.

Reviewer 2 Report

Dear Authors,

Thank you for your manuscript. Please see my comments below.

What do you mean by stating, "This study was conducted with a random sample" (line 128)? Please provide detailed information on sampling and describe the procedures of study participants recruitment. Did the authors calculate the required sample size for mediation analysis? Also, no information on research ethics is provided. The declarations section at the end of the manuscript is copied from the template and not filled in.

For CD-RISK and SCL-90 scales, Cronbach's α is extremely high (0.98-0.99). It does not indicate very good psychometric properties and rather suggests that the items were redundant or the high value is caused by a large number of items. How was this handled?

Please provide references for model fit indices (lines192-193).

Please provide the explanation of *** in Figure 2.

The authors test indirect effects/mediation within a cross-sectional design. These indirect effects are not particularly meaningful in such designs. The authors may find the below readings helpful on this topic:

https://psycnet.apa.org/doiLanding?doi=10.1037%2F1082-989X.12.1.23

https://www.tandfonline.com/doi/full/10.1080/00273171.2011.606716

The manuscript needs English revision. Some typos are found (probably I missed even more):

Abstract, lines 10-12: "The purpose of this study was to assess how coping tendency and psychological resilience affect the relationship between childhood trauma and mental health of in college students context".

Line 15: Clinical Sympsis Self-rating Scale. Please revise the scale title. In the Methods section, it is defined as Symptom Check-List 90.

Introduction, lines 70 and 83 "an mediating role".

Author Response

Response to reviewer2 comments:

What do you mean by stating, "This study was conducted with a random sample" (line 128)? Please provide detailed information on sampling and describe the procedures of study participants recruitment.

Many thanks to the reviewers for the comments. The data for this cross-sectional study were collected from a convenience sample of college students from China (Gansu Province). The participants were recruited during the second term of 2021. Of the 820 students invited to participate in the survey, 810 returned the questionnaires. In total, a sample of 740 college students from freshman to seniors, with an effective rate was 91.36%. All participants provided written informed consent prior to the study and voluntarily participated in the survey, and participants under the age of 18 were given written consent by their parents. The questionnaires were completed in a quiet classroom environment and took about 15–25 min.

In the Study Design and Participants section, we have added rich content and marked in red.

Did the authors calculate the required sample size for mediation analysis?

To calculate statistical power for moderated mediation model with current sample size (N = 740), a series of Carlo power analyses with 1,0000 replications using maximum likelihood estimation was conducted in Mplus version 8.3 to determine the power for α=0.05. Monte Carlo power analyses suggested that the sample size was sufficiently large to detect small effects (i.e., 0.20, Funder & Ozer, 2019) in multiple mediation model with power > 0.80.

In the Data analysis section, we have added rich content and marked in red.

Also, no information on research ethics is provided. The declarations section at the end of the manuscript is copied from the template and not filled in.

This study has been approved by the ethics committee of Northwest Normal University School (reference number: 20200008) . The statement at the end of the manuscript has been filled in.

In the Study Design and Participants and at the end of the manuscript section, we have added rich content and marked in red.

For CD-RISK and SCL-90 scales, Cronbach's α is extremely high (0.98-0.99). It does not indicate very good psychometric properties and rather suggests that the items were redundant or the high value is caused by a large number of items. How was this handled?

Many thanks to the reviewers for the comments. We recalculated the Cronbach's α of CD-RISK and SCL-90, which were 0.93 and 0.97 respectively. According to previous studies, the α of SCL-90 Cronbach's α is relatively high (He et al., 2022), but this does not fully explain the problem. Therefore, we also use split-half reliability test, and the result is 0.93.

He, L., Ji, X., Guo, X., & Zhang, Y. (2022). Relationship between perinatal mental health status of high-risk pregnant women and prenatal perceived stress and social support.China Journal of Health Psychology.

In the Instruments section, we have added rich content and marked in red.

Please provide references for model fit indices (lines192-193).

Many thanks to the reviewers for the comments.The modelfit was determined based on the comparativefit index (CFI) and the Tucker–Lewis index (TLI) (CFI and TLI values≥0.9 are indicative of acceptable modelfit), and the Root Mean Square Error of Approximation (RMSEA; good modelfitted indicated by values < 0.08) (Hu & Bentler, 1998).

Hu, L. T., & Bentler, P. M. (1998). Fit indices in covariance structure modeling: Sensitivity to underparameterized model misspecification. Psychological methods, 3(4), 424.

In the Data analysis and References section, we have added rich content and marked in red.

Please provide the explanation of *** in Figure 2.

Many thanks to the reviewers for the comments. Note: * means P < 0.05, * * means P < 0.01, * * * means P < 0.001.

In the Chain intermediary test of coping style and resilience section, we have added rich content and marked in red.

The authors test indirect effects/mediation within a cross-sectional design. These indirect effects are not particularly meaningful in such designs.

Thanks to the reviewers for the comments. This study was based on previous studies to explore the specific mechanisms between childhood maltreatment and mental health symptomology. But most of the articles that used cross-sectional studies mentioned in their limitations that cross-sectional studies could not draw causal relationships, so longitudinal studies were needed to explore causal relationships between variables. However, it has remained to be considered whether longitudinal studies could draw causal conclusions. Therefore, this study modified the presentation in terms of limitations and future research. The modified content is: “ First, the cross-sectional study method adopted in this study did not fully explain the causal relationship among childhood maltreatment, coping style, resilience and mental health symptomology. Future studies could use other approaches, such as longitudinal designs or experimental studies, to continue to explore this question in greater depth.”

In the limitations section, we have added extensive content and marked in red.

The manuscript needs English revision. Some typos are found (probably I missed even more): Abstract, lines 10-12: "The purpose of this study was to assess how coping tendency and psychological resilience affect the relationship between childhood trauma and mental health of in college students context".

Thanks to the reviewers for the comments. We have revised the manuscript in English. The modified content is: “The purpose of this study was to assess how coping style and resilience affect the relationship between childhood maltreatment and mental health symptomology of college students. ”

In the Abstract section, we have added extensive content and marked in red.

Line 15: Clinical Sympsis Self-rating Scale. Please revise the scale title. In the Methods section, it is defined as Symptom Check-List 90.
Thanks to the reviewers for the comments. We have unified the title of the scale. The modified content is: “ Symptom Check-List 90(SCL-90) ” .

In the Abstract section, we have added extensive content and marked in red.

Introduction, lines 70 and 83 "an mediating role".

Many thanks to the reviewers for the comments. As the introduction has been modified, the problems mentioned here have been deleted.

Reviewer 3 Report

Thank you for the opportunity to review the manuscript entitled “coping tendency and psychological resilience mediate the effect of childhood trauma on mental health”.  This is an important area of discussion, however, the manuscript could potentially be strengthened with the inclusion of additional references and some reorganizing, as well as clarification around terms. Below are some comments for consideration.

The authors must please carefully proofread the manuscript, including the abstract as there are several grammatical and sentence structure pieces that need to be revised throughout the paper.

Introduction

1.     Page 1, line 43: I would remove the phrase ‘and so on’ or consider stating what the additional main factors are.

2.     Line 44: what does contemporary college students mean?

3.     Line 45: the authors need to clarify what the living environment means- is this their home or college surroundings? Citations are also missing

4.     Line 47 I believe the authors’ meant Bronfenbrenner’s Ecological Systems Theory and not ecosystem theory

5.     Line 56-I think adverse childhood experiences is a better term that childhood trauma as there are new definitions to suggest that the experiences are what can lead to trauma. 

6.     Line 59, the types of child trauma listed are the classifications of child maltreatment, which are more limiting than adverse childhood experiences. Perhaps the authors are thinking of child maltreatment rather than child trauma/ adverse childhood experiences in general? This could also be reflected within the title

7.     Line 75 In reading the discussion on trauma and coping, I would suggest that the authors potentially reframe their introduction for flow and to comprehensively capture the international literature in the East and West. I would first have a section on evidence of associations between child maltreatment and mental health symptomology (as they’ve done but definitely include reviews and other citations as so much is published in this arena) and describe that not everyone who experiences adversity may react in the same manner. They can then mention that several theories in PTSD and resilience have proposed that coping (as they have defined) can impact resilience, and define what coping is. At this point, they can describe literature on how child maltreatment affects coping, and studies on coping and mental health. They can then mention the gap in exploring coping as a mediator of the association between child maltreatment and mental health outcomes. In looking at the literature, one recent study has looked at this relationship in Canada:

8.     https://bmcpsychiatry.biomedcentral.com/articles/10.1186/s12888-022-04001-2 .

9.     The authors can then mention why it is important to explain this relationship based on the Report on the China National Mental Health Development results to support the importance of this study. This will streamline the introduction and focus on the paper.

Methods

10.  How were participants recruited?

11.  Were participants provided with an incentive for completing the questionnaire?

12.  What were the inclusion and inclusion criteria for participating?

13.  When was the study completed?  Was the study conducted via computer or online?

14.  The authors should include a statement on receiving institutional review board approval to conduct the study.

15.  Were summary scores used for the CTQ,SCSQ, and SLC like the Connor Davidson Resilience scale?

16.  For each measure, I would include a heading that just says what variable was being explored prior to introducing the measure. For example,

a.     Child Maltreatment. The CTQ was used to XXX…

17.  Did the authors include any control variables in their analyses around demographics, relationship status, social support, substance use, academic achievement etc.? These should also be included in the paper and in the correlation analyses and SEM as they can affect resilience and coping, as well as mental health

18.  This is a grammatical comment, but the authors switch between present and past tense throughout the paper. I would consider using past tense for consistency.

Discussion

19.  As previously mentioned, there was a recent study looking at this association. However, there may also be others.

20.  In line 258, the authors use child maltreatment, which I think is a better term than childhood trauma.

21.  Line 274- avoid contractions

22.  Line 278-9: Avoid language like prove. I can understand the strong language however, I think it better to say the studies support that trauma can predict development of psychological problems. This particular paragraph seems out of place here.  The discussion would be better focused first on explaining the potential mediation effects, and then maybe stating that the remaining effect can be explained by other unmeasured factors, like neurobiological underpinnings that the authors describe.

23.  Lines 283-5- the authors cite Fan et all, which seems to be a study that is similar to the current paper. As such, this is not the first paper to examine this work as they stated in the first line of the discussion. Studies that have been conducted in this area are better placed in the introduction to highlight that it is an area that is growing, but that much work still needs to be done.

24.  288. I would not say that self- blame, fantasy and retreat are immature ways of coping.  These are complex reactions  irrespective of the age of the person who is responding to child maltreatment or other adversity.

25.  Consider breaking this paragraph around line 283 where the discussion turns to resilience.

26.  Lines 311-316- This is a very long sentence, which is a little redundant with the previous sentence.

27.  Based on my previous comments, many studies in the discussion could be used within the introduction. The discussion can be focused more on the ‘why’s behind the relationships established in the author’s work.

28.  In the limitations, the authors the demographic variables. These should be noted in the methodology.

29.  The future directions in the conclusion are rather abstract. The authors can provide more targeted directions around interventions either at the province or college level that can be used to target improved coping skills among adolescents to improve resilience and mental health.

Figures. Consider renaming mental health to mental health symptomology.

Author Response

Response to Reviewer 3 Comments

  • The authors must please carefully proofread the manuscript, including the abstract as there are several grammatical and sentence structure pieces that need to be revised throughout the paper.

Many thanks to the reviewers for the comments. After carefully proofreading of the manuscript, we revised several grammatical and sentence structures in the text. For example, "may be" in the abstract was changed to "might be", "showed" was changed to "revealed", “Childhood maltreatment indirectly affects college students' mental health symptomology through the mediating effect of coping style and resilience and the chain mediating effect of coping style and resilience. ”was changed to“Childhood maltreatment could affect college students' mental health symptomology through the chain mediating effect of coping style and resilience.” As there are many other modifications in the text, they will not be listed here.

In the whole paper, we have made the changes and marked them in green.

  • Page 1, line 43: I would remove the phrase ‘and so on’ or consider stating what the additional main factors are.

Many thanks to the reviewers for the comments. After we delete the word " and so on ", the sentence is as follows, a large number of studies had shown that interpersonal communication (Wu & Xiang, 2020), academic pressure (Zhao, 2020) and parenting style (Wen et al., 2021) were the main factors.

In line 44 of the Introduction section, we have made the changes and marked in green.

  • Line 44: what does contemporary college students mean?

Many thanks to the reviewers for the comments. We changed " contemporary college students " to "college students " after consideration.

In line 45 of the Introduction section, we have made the changes content and marked in green.

4、Line 45: the authors need to clarify what the living environment means- is this their home or college surroundings? Citations are also missing.

Many thanks to the reviewers for the comments. The aforementioned interpersonal communication, academic pressure and parental rearing patterns belong to a combination of family and school environment, so the living environment here refers to his surroundings. To avoid this misunderstanding, replace living environment with surroundings. And the corresponding references are provided.

Wu, M., Shi, S., Ge, J., & Li, Y. (2006). The Relationships Between Family Environment and Psychological Health Among Adolescents. Chinese Journal of School Health(01), 38-39.

In lines 46 and 47 of the Introduction and 543 and 544 of the Reference, we have made the changes content and marked in green.

5、Line 47 I believe the authors’ meant Bronfenbrenner’s Ecological Systems Theory and not ecosystem theory.

Thanks to the reviewers for the comments. We changed " The ecosystem theory proposed by Bronfenbrenner (1979) emphasizes that " to "Bronfenbrenner’s ecological systems theory (EST) emphasizes that ".

In lines 48 of the Introduction, we have made the changes content and marked in green.

6、Line 56-I think adverse childhood experiences is a better term that childhood trauma as there are new definitions to suggest that the experiences are what can lead to trauma. 

Thanks to the reviewers for the comments. We redefined the definition of childhood maltreatment as "Childhood maltreatment is conceptualised in terms of neglect (emotional and/or physical), and abuse (physical, emotional, and/or sexual) . "

In lines 57-59 of the Introduction, we have made the changes content and marked in green.

7、Line 59, the types of child trauma listed are the classifications of child maltreatment, which are more limiting than adverse childhood experiences. Perhaps the authors are thinking of child maltreatment rather than child trauma/ adverse childhood experiences in general? This could also be reflected within the title.

Thanks to the reviewers for the comments. According to the reviewer's opinion, we have replaced all childtraumas in the text with childhood maltreatment.

8、Line 75 In reading the discussion on trauma and coping, I would suggest that the authors potentially reframe their introduction for flow and to comprehensively capture the international literature in the East and West. I would first have a section on evidence of associations between child maltreatment and mental health symptomology (as they’ve done but definitely include reviews and other citations as so much is published in this arena) and describe that not everyone who experiences adversity may react in the same manner. They can then mention that several theories in PTSD and resilience have proposed that coping (as they have defined) can impact resilience, and define what coping is. At this point, they can describe literature on how child maltreatment affects coping, and studies on coping and mental health. They can then mention the gap in exploring coping as a mediator of the association between child maltreatment and mental health outcomes. In looking at the literature, one recent study has looked at this relationship in Canada: https://bmcpsychiatry.biomedcentral.com/articles/10.1186/s12888-022-04001-2.

Many thanks to the reviewers for the comments. According to the suggestions of reviewers, we have made corresponding adjustments to the introduction, as follows. “The above findings and theories confirmed that the experiences of childhood maltreatment led to increased mental health problems (Ainamani et al., 2021). However, not all college students who experienced adversity can develop mental health problems; instead, some adolescents develop positive coping style and eventually overcome their difficulties (Su et al., 2020). Coping style refers to the cognitive process in which individuals deal with stress and thus restore internal and external balance. And the positive consequences obtained by individuals resisting adversity are defined as resilience. Therefore, if resilience is an outcome, then coping style is the specific process of the role of resilience in the process of physical and psychological recovery of the individual(Sun YueYi, 2019). The relationship between coping style and resilience was inextricably linked from a stress-coping perspective(Clauss-Ehlers, 2008).

Coping style is defined as the thought processes and behaviors that a person sometimes engages in when he or she becomes aware of the need to manage or prevent harm and loss. One study found that childhood maltreatment was significantly and positively associated with negative coping style, and individuals who did not experienced childhood maltreatment were more likely to choose positive coping style (Zhou, 2016). Folkman and others argued that it was not the stress itself that affected an individual's physical and mental health, but the way they cope with it (Folkman et al., 1987). Furthermore, coping style was considered to be a protective factor for the physical and mental health of individuals(Tol et al., 2013). Some studies  showed that immature or negative coping style  led to psychological abnormalities among students (Pengju et al., 2018), while positive coping style was beneficial for mental health and reduce symptoms of depression and anxiety (Sun et al., 2019). Coping theory emphasized the cognitive and behavioral efforts made by individuals in the face of stress, which influence the nature and intensity of the stress response during stressful events, and thus regulate the relationship between stress and individual's mental health(Chi, 2005). A survey of African-American women found that the symptoms of psychological problems were affected by coping style after the experiences of maltreatment (Mills et al., 2018). Gökmen's study also found that coping style acted as a mediator that help mitigate the effects of maltreatment experiences on the mental health symptomatology of adolescents(Arslan, 2017). Thus, we hypothesized that coping style mediated the relationship between childhood maltreatment and mental health symptomatology.

Resilience was understood as a person's ability to always solve problems when encountering difficulties in different areas of life, and to increase one's resistance to future stress and trauma in the process of overcoming difficulties (Sisto et al., 2019). The post-traumatic resilience development model theory proposed by AGAIBI et al. (2005) emphasizes that when an individual experiences traumatic events in their early years, he or she will activate dynamic coping mechanisms to protect himself or herself, including cognitive processes about the self and the external world, mobilization and use of protective factors, and personal personality traits. Resilience may be a protective factor that mediates between childhood maltreatment and mental health symptomatology and may prevent the development of psychiatric symptoms in individuals who have experienced maltreatment (Lee et al., 2018). One study found that emotional abuse, emotional neglect, and sexual abuse were negatively associated with resilience (Kesebir et al., 2015). Individuals with higher levels of resilience reduced the occurrence of their mental health symptomatology (Xu et al., 2018). In exploring the impact of resilience on the mental health symptomatology of adolescents with experiences of childhood maltreatment, Fritz et al. (2018) found that individuals with higher levels of resilience were better able to adapt to themselves in the face of stress, thereby promoting their mental health development. More and more studies had found that resilience alleviated the depressive symptoms in individuals experiencing abuse and neglect (Lee et al., 2018). Moreover, a recent study also found that resilience mediated the relationship between childhood maltreatment and college students' mental health symptomatology (GAO et al., 2020). Zhao et al. (2021) also found that individuals who adopt positive coping style were more willing to use a variety of social resources to solve problems, so they had higher levels of resilience and better mental health (Ge et al., 2020). The above studies suggested that coping style and resilience played a role in the relationship between childhood maltreatment and mental health symptomatology among college students. However,most of the previous researches tended to focus on a single or individual factor. At an integrative level, the mechanism by which the interaction of coping style and resilience influenced the relationship between childhood maltreatment and mental health symptomatology remain poorly researched and discussed. ”

In the Introduction section, we have added extensive content and marked in green.

9、The authors can then mention why it is important to explain this relationship based on the Report on the China National Mental Health Development results to support the importance of this study. This will streamline the introduction and focus on the paper.

Many thanks to the reviewers for the comments. We have revised it according to the comments of reviewers, and the results are as follows. Based on the above findings and theories, this study aimed to investigate the effects of college students' childhood maltreatment on mental health symptomology.

In lines 124-125 of the Introduction section, we have added extensive content and marked in green.

10、How were participants recruited?

Many thanks to the reviewers for the comments. The data for this cross-sectional study were collected from a convenience sample of college students from China (Gansu Province). The participants were recruited during the second term of 2021. Of the 820 students invited to participate in the survey, 810 returned the questionnaires.

Because of the same opinion with the first reviewer, in lines 141-144 of the Study Design and Participants section, we have added extensive content and marked in red.

11、Were participants provided with an incentive for completing the questionnaire?

Many thanks to the reviewers for the comments. All participants provided written informed consent prior to the study and voluntarily participated in the survey, and participants under the age of 18 were given written consent by their parents.

Because of the same opinion as with the first reviewer, in lines 149-151 of the Study Design and Participants section, we have added extensive content and marked in red.

12、What were the inclusion and inclusion criteria for participating?

Many thanks to the reviewers for the comments. Students from the School of Psychology and Education of a university in Gansu Province. The following situations will be excluded, where 10 or more questions in a row selected the same answer, the answers presented in a particular pattern (such as 121212, etc.) and 5 or more questions were missed.

In lines 144-147 of the Study Design and Participants section, we have added extensive content and marked in green.

13、When was the study completed?  Was the study conducted via computer or online?

Many thanks to the reviewers for the comments. The participants were recruited during the second term of 2021. The questionnaires were completed in a quiet classroom environment and took about 15–25 min.

Because of the same opinion with the first reviewer, in lines 142-143 and 155-156 of the Study Design and Participants section, we have added extensive content and marked in red.

14、The authors should include a statement on receiving institutional review board approval to conduct the study.

Many thanks to the reviewers for the comments. This study has been approved by the ethics committee of Northwest Normal University School. 

Because of the same opinion with the first reviewer, in lines 156-157 of the Study Design and Participants section, we have added extensive content and marked in red.

15、Were summary scores used for the CTQ,SCSQ, and SLC like the Connor Davidson Resilience scale?

Many thanks to the reviewers for the comments. These scales have different scoring methods. The total score of CTQ was the sum of 25 items except three valid items. The total score of SCSQ is positive dimension standard score minus negative dimension standard score. The total score of SCL-90 is the sum of 90 items.

In the Instruments section, we have added extensive content and marked in green.

  • For each measure, I would include a heading that just says what variable was being explored prior to introducing the measure. For example, a. Child Maltreatment.

Many thanks to the reviewers for the comments. After consideration, we finally adopted the comments of the reviewers.

In the Instruments section, we have added extensive content and marked in green.

  • Did the authors include any control variables in their analyses around demographics, relationship status, social support, substance use, academic achievement etc.? These should also be included in the paper and in the correlation analyses and SEM as they can affect resilience and coping, as well as mental health.

Many thanks to the reviewers for the comments. We controlled the gender of the participants (0 = male, 1 =female ) because it was reported that gender was related to the mental health of individuals(e.g., Cauce et al., 2000). 

Because of the same opinion with the second reviewer, in lines 204-207 of the Instruments section, we have added extensive content and marked in blue.

  • This is a grammatical comment, but the authors switch between present and past tense throughout the paper. I would consider using past tense for consistency.

Thanks to the reviewers for the comments. We changed the whole article into the past tense.

  • As previously mentioned, there was a recent study looking at this association. However, there may also be others.

Thanks to the reviewers for the comments. On the one hand, there are more and more researches on the mechanisms between childhood maltreatment and mental health symptomatology, which shows that this problem has attracted the attention of researchers. This research is similar to previous researches, but it is not exactly identical. On the basis of previous studies, this study has made some contributions to enrich this research. On the other hand, due to different cultural backgrounds, there may be some differences in the specific mechanisms of action. We hope that starting from different cultural backgrounds, we can make our contribution to the establishment of a universal mechanism.

  • In line 258, the authors use child maltreatment, which I think is a better term than childhood trauma.

Thanks to the reviewers for the comments. We have chosen to use the word childhood maltreatment.

  • Line 274- avoid contractions

Thanks to the reviewers for the comments. We changed "can't" to "can not".

In 316 lines of the Discussion section, we have made changes and marked in green.

  • Line 278-9: Avoid language like prove. I can understand the strong language however, I think it better to say the studies support that trauma can predict development of psychological problems. This particular paragraph seems out of place here.  The discussion would be better focused first on explaining the potential mediation effects, and then maybe stating that the remaining effect can be explained by other unmeasured factors, like neurobiological underpinnings that the authors describe.

Thanks to the reviewers for the comments. We modify this paragraph to read “Consistent with prior studies (Zhang et al., 2021), childhood maltreatment positively predicted college students' mental health symptomology, indicating that college students who had adverse experiences in childhood had a higher chance of having mental health problems. This may be due to the lack of communication between college students with childhood maltreatmenttic experiences and their caregivers in childhood,  did not feel the care and love they deserved, and did not form the correct and healthy value orientation during their growth process, resulting in their failure to form a healthy psychology (Liu et al., 2019). The studies support that trauma could predict development of psychological problems.” And using neurobiological basis to explain the remaining effects can be explained by other unmeasured factors.

In 306-314 and 347-356 lines of the Discussion section, we have made changes and marked in green.

  • Lines 283-5- the authors cite Fan et all, which seems to be a study that is similar to the current paper. As such, this is not the first paper to examine this work as they stated in the first line of the discussion. Studies that have been conducted in this area are better placed in the introduction to highlight that it is an area that is growing, but that much work still needs to be done.

Thanks to the reviewers for the comments. We have revised this sentence and used new references. “ First of all, childhood maltreatment negatively predicted coping style, while coping style positively predicted college students' mental health symptomology (Ma et al., 2018). ”

Ma, S., Wan, Y., Zhang, S., Xu, S., Liu, W., Xu, L., ... & Tao, F. (2018). Mediating effect of psychological symptoms, coping styles and impulsiveness on the relationship between childhood abuses and non-suicidal self-injuries among middle school students. Journal of Hygiene Research, (04),530–535.

In 318-319 lines of the Discussion section, we have made changes and marked in green.

  • Line 288. I would not say that self- blame, fantasy and retreat are immature ways of coping.  These are complex reactions  irrespective of the age of the person who is responding to child maltreatment or other adversity.

Thanks to the reviewers for the comments. Let's replaced "immature" with "some".

In 323 lines of the Discussion section, we have made changes and marked in green.

  • Consider breaking this paragraph around line 283 where the discussion turns to resilience.

Thanks to the reviewers for the comments. We breaked this paragraph around line 328 where the discussion turns to resilience.

  • Lines 311-316- This is a very long sentence, which is a little redundant with the previous sentence.

Thanks to the reviewers for the comments. Let's changed this sentence to " This was consistent with the research results of Yang et al. (2017). "

In 347-348 lines of the Discussion section, we have made changes and marked in green.

  • Based on my previous comments, many studies in the discussion could be used within the introduction. The discussion can be focused more on the ‘why’s behind the relationships established in the author’s work.

Many thanks to the reviewers for the comments. In response to this question raised by the reviewers, we have made new changes to the Discussion section and marked it with green.

  • In the limitations, the authors the demographic variables. These should be noted in the methodology.

Many thanks to the reviewers for the comments. All of the data on the main variables of interest (i.e., Childhood Maltreatment, Coping Style, Resilience and Mental Health Symptomology) used in this study were collected on the same day. Collect information such as gender, grade, and school nature through demographic survey. 

Because of the same opinion with the second reviewer, in lines 151-155 of the Study Design and Participants section, we have added extensive content and marked in blue.

  • The future directions in the conclusion are rather abstract. The authors can provide more targeted directions around interventions either at the province or college level that can be used to target improved coping skills among adolescents to improve resilience and mental health.

Many thanks to the reviewers for the comments. We propose a more targeted approach. First, for the nurturers of children to carry out publicity and education in the form of family activities, not only to pay attention to the physical development of children, but also to create a warm and safe family atmosphere and emotional support to give more attention to children, and pay attention to the communication and exchange with children. Secondly, for example, the mental health week is used to play short animation videos about coping tendencies, and different coping styles bring different results to guide them to choose more appropriate coping tendencies when they encounter things. Develop a series of small activities centered on the ABC theory of emotion (trivia role-play, etc.) to change their irrational beliefs and improve their mental resilience.

In the Conclusion section, we have added extensive content and marked in green.

  • Consider renaming mental health to mental health symptomology.

Many thanks to the reviewers for the comments. We renamed mental health as mental health symptomatology.

Round 2

Reviewer 2 Report

Dear Authors,

Thank you for the improved version of your manuscript. Good job!